# Self-Reported Medication Adherence Measured with Morisky Scales in Rare Disease Patients: A Systematic Review and Meta-Analysis

**DOI:** 10.3390/healthcare11111609

**Published:** 2023-05-31

**Authors:** Ana María García-Muñoz, Desirée Victoria-Montesinos, Begoña Cerdá, Pura Ballester, Eloisa María de Velasco, Pilar Zafrilla

**Affiliations:** Faculty of Pharmacy and Nutrition, Universidad Católica San Antonio de Murcia (UCAM), Campus de los Jerónimos, 30107 Murcia, Spain

**Keywords:** rare diseases, treatment adherence, Morisky adherence scale, meta-analysis, systematic review

## Abstract

Background: The visibility of Rare Diseases is a new challenge for society. These diseases are numerous, heterogeneous in nature and distribution, characterized by a high mortality rate but low prevalence, and usually presenting a severe evolution. Adherence to medication studies in rare diseases are uncommon, due to treatment scarcity. Objectives: The main purpose of this study is to do a meta-analysis, evaluating the level of adherence to medication in the most prevalent rare diseases. Methods: This work is a systematic review, and meta-analysis was registered in the International Prospective Register of Systematic Reviews (PROSPERO) (Registration number: CRD42022372843) and conducted according to the Preferred Reporting Items for Systematic Reviews and Meta-Analyses (PRISMA) statement. Adherence to treatment in this systematic review and meta-analysis was collected from all studies included, based on the crude numerators and denominators reported, using either the Morisky Medication Adherence Scale 4 or -8. Results: A total of 54 records were identified through database searches, or after screening relevant manuscripts’ references. Finally, 18 studies were included in this systematic review and meta-analysis. A total of 1559 participants (54.18% women) aged less than 84 years old were included. Twelve studies used the MMAS-8. In 8 of them, they established the level of adherence to treatment in three categories (low, medium, and high), with the mean prevalence in each of them being 41.4%, 30.4%, and 28.2%, respectively. Conclusions: The results observed in adherence to treatment in patients with rare diseases show great variability, due to the different aspects involved in the greater or lesser applicability of the medication.

## 1. Introduction

In recent years, a new challenge has arisen for society: the visibility of Rare Diseases (RDs) at a multidisciplinary level, integrating health, education, and social awareness [1]. Humanizing RD is one of the Sustainable Development Goals (SDGs) included by the United Nations in the 2030 Agenda [2].

These diseases are numerous, and heterogeneous in nature and distribution, with a high mortality rate [3]. Usually, they involve a severe evolution of the condition, with multiple motor, sensory, and cognitive impairments, often presenting a high level of clinical complexity, making their recognition, diagnosis, and treatment difficult [4].

The actual number of people living with RDs is difficult to be determined. In several parts of the world, the rarity criterion is determined by regulations intended to encourage industry investment in RDs’ drug discovery, or marked by the notion of patients’ prevalence in a region [5]. The accepted prevalence of RD in the United States is 1 in 1500 people; in Australia it is stipulated at 1 in 10,000 [6]; and 1 in 2500 citizens in Japan [7]. In Spain, we adhered to the criterion established by the European Union, considering a disease as rare when it affects no more than 1 person in 2000 [3,4,8].

Epidemiological studies are difficult, as most research addressing incidence comes from national, often local, registries on specific diseases or groups of diseases [3]. Around 6000 and 8000 RDs affect 30 million people in the European Union [9]. The ISPOR Rare Disease Special Interest Group published a global systematic review, stating that the average prevalence of a RD was between 40 and 50 cases/100,000 people, and, despite all the variations, a coordinated effort is needed to standardize objective criteria and avoid qualitative descriptors [7]. However, accuracy of RD prevalence is crucial. Knowing that, we could better determine patients’ health care system unmet needs, improve condition management, and estimate the number of individuals benefiting from novel drug development, existing therapies for RDs, or ongoing clinical trials.

Adherence, understood as patient compliance of medical recommendations [10], is a problematic phenomenon well-studied in highly prevalent chronic pathologies, such as diabetes and hypertension, or in moderately prevalent diseases, such as HIV [11]. Numerous factors influence adherence to drug therapy, some related to the patient and clinicians, and others to the medications (frequency of administration, length of treatment, or tolerability). Optimal communication between all actors (prescribers, pharmacists, and patients) influences treatment adherence [12].

Lack of adherence is a serious public health problem affecting healthcare systems worldwide, especially when scarce medications are available, as is the RD scenario [13]. In a recently published manuscript of RDs, a cross-sectional investigation conducted in 139 Wilson disease patients at the National Reference Center for Wilson’s Disease (CRMR) revealed that, as in many chronic diseases, patients were weakly adherent [14]. However, a Polish piece of research in the same condition described that 74.1% of symptomatic patients were adherent to the prescribed medication [15]. In adopting a patient-centered care approach, pharmacists could play an important role identifying and resolving medication-related problems and contributing to improve treatment adherence, impacting on healthcare system sustainability [16].

To the best of our knowledge, adherence studies in patients with RDs are unusual [15], as most of them lack approved and effective therapies. The main purpose of this study is to analyze, with the methodology of meta-analysis, the published works about adherence to medication of the most prevalent rare diseases.

## 2. Materials and Methods

This systematic review and meta-analysis was registered in the International Prospective Register of Systematic Reviews (PROSPERO, Registration number: CRD42022372843) and conducted according to the Preferred Reporting Items for Systematic Reviews and Meta-Analyses (PRISMA) statement [17].

### 2.1. Eligibility Criteria

The following inclusion criteria were established: (a) Participants: the participants had a rare disease diagnosis; (b) Outcome: treatment adherence measured by Morisky Medication Adherence Scale (MMAS, 4- or 8- items); and (c) Study design: no restriction, with the exception of systematic reviews, and/or meta-analyses, qualitative and case studies. Studies were limited to those published in English or Spanish. The exclusion criteria included studies: (a) With participants without a RD; (b) With adherence measured with a scale different to the MMAS; (c) Based on data from the same survey/study; (d) That were not randomized controlled trials, cross-sectional, or longitudinal studies, specifically excluding case-control studies, cohort studies, case reports, case series, qualitative studies, systematic reviews, meta-analyses, experimental animal studies, *in vitro* studies, and expert opinions or consensus statements; (e) In a language other than English or Spanish.

### 2.2. Information Sources and Search Strategy

Two researchers (AMG-M and DV-M) systematically searched the PubMed, Scopus, Web of Science, and Cochrane Database of Systematic Reviews databases, with a date limit from January 2005 to November 2022. Studies were identified via the following search terms: (a) “Morisky Medication Adherence Scale”, “MMAS-4”, “MMAS-8”, “Morisky Green Levine”, “Morisky Green Levine Medication Adherence Scale”, “Medication Adherence Questionnaire”; (b) “Rare disease”, “Cystic Fibrosis”, “Hemophilia A”, “Hemophilia B”, “Idiopathic Pulmonary Fibrosis”, “Myasthenia Gravis”, “Sickle Cell Disease”, “Primary biliary cholangitis”, “Fabry disease”, “Pulmonary arterial hypertension”, “Wilson’s disease”, “Narcolepsy”. The search terms were adapted for each database, in combination with database-specific filters (provided in Appendix A). In the search, we used those rare diseases with the highest prevalence and in which the pharmaceutical industry invests the most money [18,19,20].

### 2.3. Selection Process

After identifying eligible studies, Mendeley (Version for Windows 10; Elsevier, Amsterdam, Netherlands) was used to remove the duplicates. Two members of the research team (A.M.G.-M. and D.V.-M.) conducted the selection process independently, and screened all titles and abstracts to identify potentially relevant articles for further review in the full-text phase. A third researcher (E.M.-G.) participated in resolving discrepancies.

### 2.4. Data Items

Study details, such as sample size, country, study design, and type of medication used, were extracted. The proportion of participants with adherence to treatment was extracted by one researcher (D.V.-M.); meanwhile, another researcher (A.M.G.-M.) checked the data for accuracy. In case of a discrepancy between these two researchers, a third researcher (E.M.-G.) reviewed the information. 

### 2.5. Risk of Bias Assessment

Two researchers (D.V.-M. and A.M.-G.) independently assessed the risk of study bias of the included studies. The assessment of the risk of bias was carried out using a specific tool for prevalence and proportion studies [21]. This comprehensive tool evaluates a wide range of potential biases, examining 10 items that cover various aspects of internal and external validity. These include, but are not limited to, sample representativeness, sample size, non-respondents, data collection method, case definition, measurement tool validity and reliability, and statistical analysis, thus offering a broad evaluation of the potential biases in the studies we reviewed. Each item is classified with the answer “yes” (low risk) or “no” (high risk), with a score of 0 and 1 point, respectively. Depending on the score, the study will be classified as “low risk of bias” (scores of 0–3), “moderate risk of bias” (scores of 4–6), or “high risk of bias” (scores of 7–9).

### 2.6. Outcome Measures

Adherence to treatment in this systematic review and meta-analysis was collected from all studies included, based on the crude numerators and denominators reported using either MMAS-4 [22] or -8 [23]. In those studies that used the MMAS-4 scale, participants were divided as “with adherence to treatment” and “non-adherence to treatment”. On the other hand, those that used the MMAS-8 scale classified the sample into “high adherence” (8), “medium adherence” (6–8), and “low adherence” (<6). For comparison purposes, those subjects with “high adherence” and “medium adherence” in MMAS-8 were reclassified as “with adherence to treatment”. 

### 2.7. Synthesis Methods

Using Stata (Version 16.1; StataCorp., College Station, TX, USA) and the metaprop package [24], the proportion of multiple studies was pooled by applying a random-effects, using the DerSimonian and Laird method and a general linear mixed model (GLMM) [25]. The exact, or Clopper-Pearson, method was used to establish 95% confidence intervals (95% CIs) for the proportions from the selected individual studies [26], and a Freeman-Tukey transformation was used to normalize the results, before calculating the pooled proportion [27].The estimated effect was also performed with different transformation methods, such as the arcsine and logit transformations [27,28]. Intragroup heterogeneity of pooled proportions was also calculated using the *I*^2^ statistic and its p-value. Small-study effects and publication bias were examined using Egger’s test and funnel plots.

Sub-group analyses were conducted by type of disease, age group, study site, and Risk of Bias score. For the subgroup according to age, the studies were divided into two groups according to the mean age: “children, adolescents and young adults” with an age between 0 and 24 years, and another group formed by those subjects with a mean age above 24 years old, called “adults and seniors”. In addition, random-effects meta-regression analyses, using the method of moments, were estimated to independently assess whether treatment adherence differed by mean age, year of publication, or quality score of the studies.

## 3. Results

### 3.1. Study Selection

A total of 54 records were identified through database searches and in other articles’ bibliographies (Figure 1). After screening for duplicates, 27 records remained. Finally, 25 studies were obtained for full-text review. Of those studies, 7 were excluded to avoid redundancy, as were extracting data from the same study [29,30,31] or not showing adherence to treatment data [32,33,34,35]. Finally, 18 studies [14,36,37,38,39,40,41,42,43,44,45,46,47,48,49,50,51,52] were included in this systematic review and meta-analysis.

### 3.2. Study Characteristics

Table 1 summarized the main characteristics of the 18 included studies. A total of 1559 participants (54.18% women) aged 0−83 years were included in this systematic review and meta-analysis. 

Based on the type of measurement used, there were 12 studies that used the MMAS-8 [14,36,37,38,41,43,45,46,47,48,49,50]. In 8 of them [14,36,38,43,46,47,48,50], they established the level of adherence to treatment in three ranges (low, medium, and high), with the mean prevalence in each of them being 41.4%, 30.4%, and 28.2%, respectively. On the other hand, 6 studies used the MMAS-4 [39,40,42,44,51,52]. The mean adherence reported for this type of scale was 52.9%. According to sex, 15 studies reported the overall proportion of adherence in both men and women [14,37,38,39,40,41,43,44,45,46,47,48,49,51,52], and 3 studies only included one sex [36,42,50] (i.e., only women).

Regarding the type of rare disease, there were 6 studies in which the participants had sickle cell disease (SCD) [38,39,40,41,45,49], also including myelodysplastic syndromes and β-thalassemia in one of them [39]. Four studies included subjects with myasthenia gravis (MG) [37,46,51,52], and two studies included subjects with congenital hypogonadotropic hypogonadism [36,50]. The rest of the subjects in the different studies had different RDs, such as congenital Wilson’s disease [14], Fabry disease [44], cystic fibrosis [43], hemophilia A [42], pulmonary arterial hypertension [48], and amyotrophic lateral sclerosis [47].

In terms of geographical regions, 9 different countries were identified, including 4 regions in Europe [14,39,42,43,47,48], 2 in South America [37,52], 1 in Asia [44,51], 1 in North America [38,40,45,46], and 1 in Africa [41,49]. Two studies did not specify the country, since it was performed via online, on Facebook and other social media [36,50].

### 3.3. Risk of Study Bias

All the studies showed a low risk of bias, presenting scores between zero and three points. Two studies showed a total of three points [38,40]. The main sources of bias were related to national sample representation [37,38,40,41,43,44,45,47,48,49,50,51,52]. A summary of the risk of bias is presented in the Table 2.

### 3.4. Results of Syntheses

#### Treatment Adherence

Figure 2 shows that the overall proportion of treatment adherence was 57.14% (95% CI: 44.09% to 69.73%; *p* < 0.001, *I*^2^ = 95.54%; GLMM: 54.95% (95% CI: 32.30% to 77.60%; *p* < 0.001). The results obtained with the logit transformation and the arcsine transformation were similar (logit transformation: 53.4% (95% CI: 43.4% to 63.5%; *p* < 0.001, *I*^2^ = 90.37%); arcsine transformation: 55.7% (95% CI: 44.0% to 67.4%; *p* < 0.001, *I*^2^ = 88.90%). The Egger’s test showed no significant differences for any of the variables analyzed in the meta-analysis (*p* = 0.19), indicating an absence of publication bias. However, visual assessment using the funnel plot suggests that publication bias exists, although the Egger’s test result is not statistically significant (provided in Appendix A).

A subgroup analysis was performed for those diseases in which two or more studies were found that measured adherence using MMAS-4 or MMAS-8. In this analysis, considerable variation in adherence was observed, depending on the type of disease, with the highest adherence in myasthenia gravis (63.19%, 95% CI: 39.16 to 84.28), followed by Sickle Cell Disease (56.91%, 95% CI: 30.61 to 81.38), and congenital hypogonadotropic hypogonadism (41.01%, CI: 33.35 to 48.89).

Figure 3 shows the subgroup analysis in relation to age group. The overall proportion was slightly higher in the seniors’ group (60.90%, 95% CI: 46.39% to 74.52%; *p* < 0.001) than in the youth group (49.15%, 95% CI: 23.09% to 75.45%; *p* < 0.001).

In the subgroup analysis according to the study site, it was observed that the studies conducted in Europe showed a greater overall proportion (68.38%, 95% CI: 49.34% to 84.74%; *p* < 0.001), whereas, in studies conducted online, the overall proportion was lower (41.01%, 95% CI: 33.35% to 48.89%).

According to the risk of bias, it was observed that those studies with a lower score on this scale had higher overall proportions: Total score equal to 0: 61.65%, 95% CI: 41.98% to 79.54%, and Total score equal to 1: 70.86%, 95% CI: 46.52% to 90.22%.

Table 2 shows the random-effects meta-regression models of mean age, year of publication, and quality score of the studies, with respect to overall treatment adherence. Treatment adherence was not associated with these parameters (*p* > 0.05 in all variables). The random-effects meta-regression models of mean age, with regards to the overall treatment adherence, are shown in Figure 4. 

## 4. Discussion

To the best of our knowledge, this is the first meta-analysis that has comprehensively examined the overall proportion of treatment adherence in different RDs. The main findings of this study are as follows: (a) a total of 57.14% of 1559 participants from 9 countries have adequate adherence to the treatment prescribed for their specific disease; (b) no significant differences were found in medication adherence considering the age group; (c) adherence was not associated with mean age. There was a large variation in adherence to treatment, depending on the type of disease and medication used.

In all the RD included in this meta-analysis, different types of treatment have been studied for years to mitigate or reduce side effects, even though they are not entirely effective [53]. Adherence to drug treatment has been found to be essential in all diseases, including RDs. However, the difficulties presented by this type of patient, both physical and psychological, may impact adherence rate being not as high as it should be. In patients with RDs, poor adherence or treatment interruption will lead to a worsening of the disease itself [54].

Of the diseases analyzed in this meta-analysis, SCD has shown the highest adherence. It is a rare genetic disease, caused by a mutation of the beta-chain of hemoglobin, producing an alteration of the erythrocyte shape, and resulting in a large formation of cell aggregates, which can eventually lead to hypoxia, hypercoagulability, increased platelet activation, and increased neutrophil adhesiveness. Adebiyi and collaborators described that factors such as climate, fetal hemoglobin levels, and even certain infections may play a definitive role in the manifestation of this disease [55]. There are multiple treatments for this disease; however, in the studies used in this meta-analysis, the use of hydroxyurea has predominated [41,45,49].

The mean adherence measured with MMAS-8 in the six included studies [38,39,40,41,45,49] was 56.91%. In a systematic review and meta-analysis of 14 studies performed by Loiselle et al. [56], with a sample of 921 persons, results similar to those of this meta-analysis were observed, showing an overall adherence to pharmacological treatment, measured by various methods, of 50% in patients with SCD. In a systematic review by Walsh et al. [57], they observed that self-reported measures of adherence, such as the MMAS-4 or MMAS-8, tend to have a higher compliance rate (48–89%) than those measured by objective methods, such as urinalysis. These authors observed a higher adherence rate in patients who used hydroxyurea as a drug, a result that coincides with that observed in our meta-analysis, which shows an adherence rate of 57.25% in patients treated with hydroxyurea. Increasing adherence is important, since this RD primarily affects children. One solution may be setting preventive clinic visits. A systematic review, conducted in 2010 by Dean and collaborators [58], concluded that behavioral and educational interventions are effective in achieving greater adherence, and help parents and caregivers to correctly give medication to their children.

On the other hand, this systematic review and meta-analysis included a total of 4 studies [37,46,51,52] with patients suffering from MG. MG is an autoimmune disease of neuromuscular origin, characterized by different symptoms that depend on the degree of involvement of the patient’s striated muscle [59]. Current treatment is based on generalized and nonspecific immunosuppression. It is a disease that responds well to pharmacological treatment, so adherence is essential for this type of patient.

The heterogeneity of MG results in multiple therapeutic approaches, depending on the subtype. Therefore, Guptill et al. discussed that clinical trials to find an effective treatment must be performed in a population that is as homogeneous as possible, which is sometimes a complicated objective, due to the fact that it is a RD [60]. In this meta-analysis, adherence of 63.19% was observed, measured by MMAS-4 and MMAS-8; the value was higher than those observed in patients with SCD. These differences may be due to the type of treatment used. In patients with MG, chronic treatment is carried out with easily applied immunosuppressive drugs, such as corticosteroids, which produce a remission or decrease in symptoms in 70% of patients, as mentioned in the work of Alhaidar and colleagues [61]. These are usually used in combination with other immunosuppressive drugs, such as cyclosporine or methotrexate, to reduce the dose of corticosteroids, achieving an even greater improvement in the patient’s symptoms [60].

Other rare diseases analyzed in this meta-analysis have very different adherence to treatment, ranging from 16.42% to 88.21%. As mentioned above, this review has included the most common rare diseases studied by the pharmaceutical [19,20]. Adherence to pharmacological treatment is one of the major concerns of health professionals (physicians, pharmacists), due to the importance of reducing the symptoms of this type of disease.

Congenital hypogonadotropic hypogonadism [62], pulmonary arterial hypertension [63], cystic fibrosis [64], Wilson’s disease [65], etc. are diseases whose drugs can help patients maintain their quality of life through chronic treatment. However, adherence to treatment will depend on different aspects, some related to the patient (lack of motivation, depression, denial, cognitive deterioration, etc.), and others related to the pharmacological treatment (complexity, side effects, time, etc.) [66].

The factors that can increase adherence to pharmacological treatment are the support of both family and health professionals, the ease of taking the drugs, the benefits perceived by the patient, and the establishment of a routine for taking the medication. Another factor that can favor increased adherence in patients with this type of disease is the dispensing of these drugs in community pharmacies and primary care centers [64].

This systematic review and meta-analysis has several limitations. First, despite performing an exhaustive search for different rare diseases, it is possible that some relevant articles were not included. Second, unpublished literature has not been included. Third, the studies included in the article used self-report questionnaires, which may result in a “social desirability and recall bias”. In addition, the gray literature was not used in this review, which may lead to loss of information. Finally, there are very few studies using a sample with a rare disease, and only the MMAS was used to measure adherence, so the results should not be generalized and should be interpreted with caution.

## 5. Conclusions

The results observed in adherence to treatment in patients with rare diseases had great variability, due to the different aspects involved in the greater or lesser applicability of the treatment. Low adherence is a heterogeneous and multifactorial problem that requires not only health professionals, as providers or pharmacists, but also the intervention of psychologists and a multidisciplinary team, including the family. In addition, such treatment must be established individually, according to individuals’ characteristics.

## Figures and Tables

**Figure 1 healthcare-11-01609-f001:**
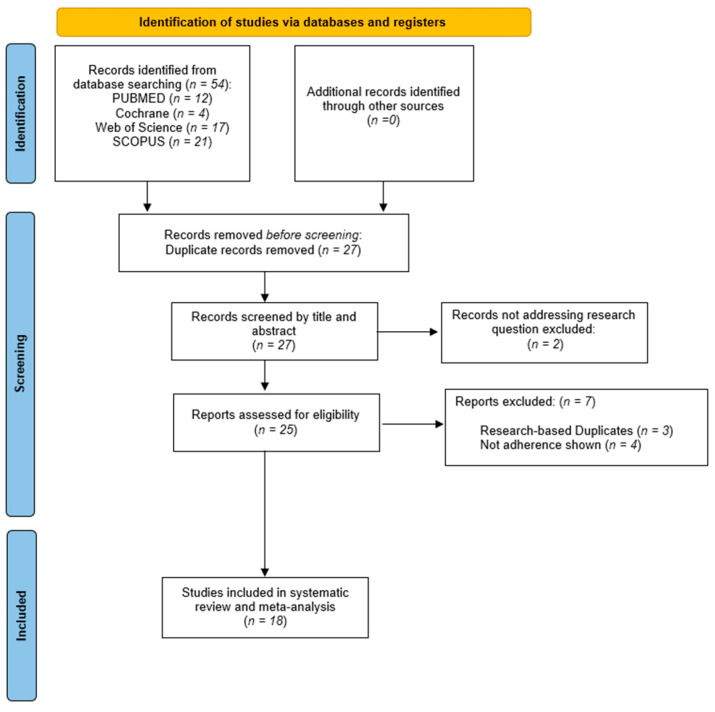
Identification of studies via databases and registers.

**Figure 2 healthcare-11-01609-f002:**
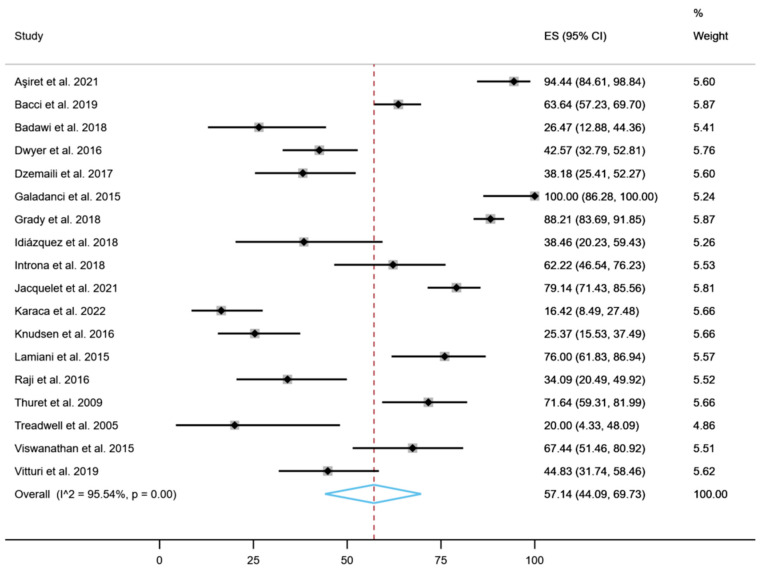
Overall proportion of treatment adherence (%) [14,36,37,38,39,40,41,42,43,44,45,46,47,48,49,50,51,52].

**Figure 3 healthcare-11-01609-f003:**
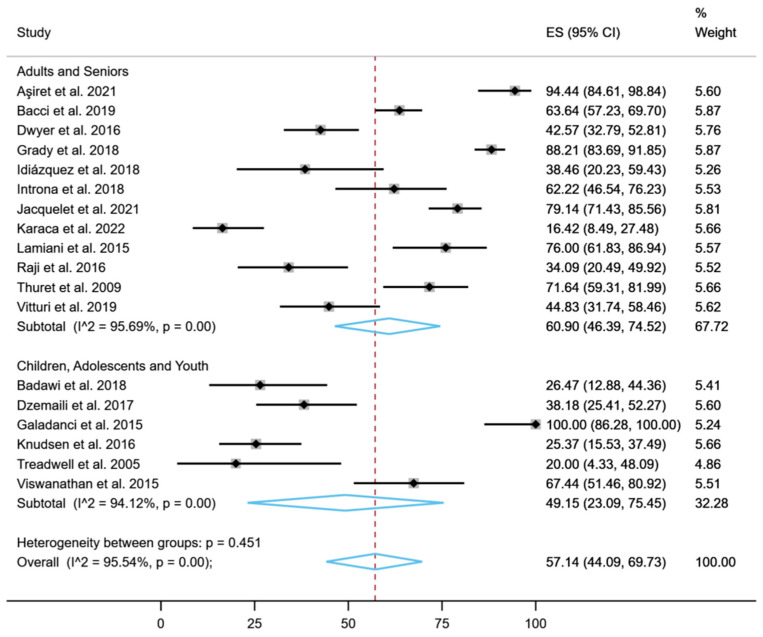
Subgroup analysis in relation to age group for treatment adherence (%) [14,36,37,38,39,40,41,42,43,44,45,46,47,48,49,50,51,52].

**Figure 4 healthcare-11-01609-f004:**
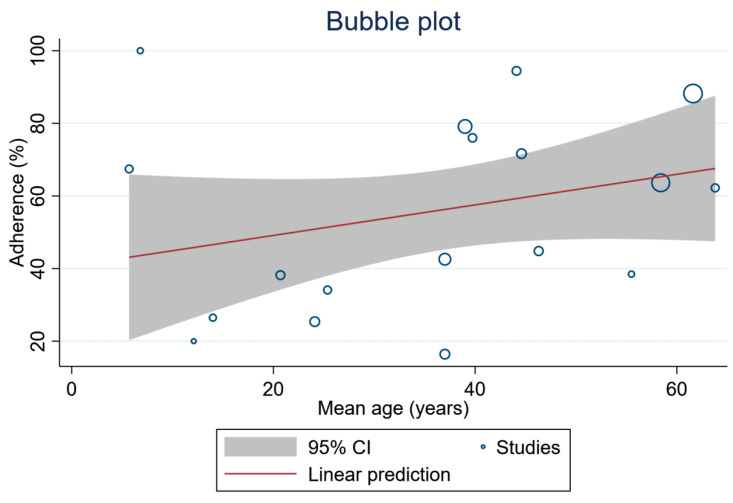
Random-effects meta-regression models of mean age.

**Table 1 healthcare-11-01609-t001:** Characteristics of the studies included (N = 18).

Reference	Year	Country	Study Design	Total (n)	Women (%)	Age (Mean)	Rare Disease	Tool	Therapy	Adherence (%)
Aşiret et al. [51]	2021	Turkey	Cross-sectional	54	64.8	44.1	Myasthenia gravis	MMAS-4	Cholinesterase inhibitors, Glucocorticoids (methyl prednisolone, prednisone vb.), Ciclosporin	94.5
Bacci et al. [46]	2019	USA	Cross-sectional	242	64.0	58.4	Myasthenia gravis	MMAS-8	Intravenous immunoglobulin	63.7
Badawi et al. [45]	2018	USA	Cross-sectional	34	41.0	14.0	Sickle cell disease	MMAS-8	Hydroxyurea	26.0
Dwyer et al. [50]	2016	Online	Cross-sectional	101	0	37.0	Congenital hypogonadotropic hypogonadism	MMAS-8	Testosterone replacement therapy or fertility-inducing treatment via exogenous	42.6
Dzemaili et al. [36]	2017	Social media sites	Cross-sectional	55	100	20.7	Congenital hypogonadotropic hypogonadism	MMAS-8	Hormone replacement	38.2
Galadanci et al. [49]	2015	Nigeria	Randomized clinical trial	25	52.0	6.8	Sickel cell disease	MMAS-8	Hydroxyurea	100
Grady et al. [48]	2018	United Kingdom	Cross-sectional	263	70.6	61.6	Pulmonary arterial hypertension	MMAS-8	Ambrisentan, bosentan, sildenafil, tadalafil, iloprost, epoprostenol, ERA + PDE5i, Iloprost (nebulized) + PDE5i, IV/SC Prostanoid + ERA, IV/SC Prostanoid + PDE5i, Trial drug + ERA +/− PDE5i	88.2
Idiázquez et al. [52]	2018	Chile	Cross-sectional	26	57.7	55.5	Myasthenia gravis	MMAS-4	Cholinesterase inhibitors or immunosuppressors	38.5
Introna et al. [47]	2018	Italy	Cross-sectional	45	40.0	63.8	Amyotrophic lateral sclerosis	MMAS-8	Riluzole tablet or oral suspension of riluzole	62.2
Jacquelet et al. [14]	2021	France	Cross-sectional	139	50.4	39.0	Wilson’s disease	MMAS-8	D-Penicillamine, Trientine 2HCl, Zinc acetate, Zinc sulfate and Zinc sulfate	79.1
Karaca et al. [44]	2022	Turkey	Cross-sectional	67	52.2	37.0	Fabry disease	MMAS-4	Enzyme replacement therapy	16.4
Knudsen et al. [43]	2016	Denmark	Cross-sectional	67	59.0	24.1	Cystic fibrosis	MMAS-8	NR	25.8
Lamiani et al. [42]	2015	Italy	Cross-sectional	50	0	39.7	Hemophilia A	MMAS-4	On-demand and prophylaxis	76.0
Raji et al. [41]	2016	Nigeria	Cross-sectional	205	86.3	25.4	Sickle cell disease	MMAS-8	Hydroxyurea	34.1
Thuret et al. [39]	2009	France	Cross-sectional	70	54.0	44.6	Sickle cell disease myelodysplastic syndromes and β-thalassemia	MMAS-4	DFO, deferiprone, deferiprone + DFO or deferasirox	72.0
Treadwell et al. [40]	2005	USA	Cross-sectional	15	53.5	12.1	Sickle cell disease	MMAS-4	Chelation therapy	20.0
Viswanathan et al. [38]	2015	USA	Cross-sectional	43	48.8	5.7	Sickle cell disease	MMAS-8	Hydroxyurea and Penicillin	69.0
Vitturi et al. [37]	2020	Brazil	Cross-sectional	58	81.0	46.3	Myasthenia gravis	MMAS-8	NR	44.8

DFO: Deferoxamine; ERA: endothelin antagonist; MMAS-4: Morisky Medication Adherence Scale-4; MMAS-8: Morisky Medication Adherence Scale-8; MDS: myelodysplastic syndromes; NR: Not reported; PDE5i: phosphodiesterase type-5 inhibitor; SCD: Sickle cell disease.

**Table 2 healthcare-11-01609-t002:** Characteristics of the meta-regression model.

Variable	Coefficient	Lower Limit Confidence Interval	Upper Limit Confidence Interval	*p*-Value
Age mean	0.32	−0.44	1.08	0.403
Year of publication	−0.34	−3.78	3.10	0.845
Quality score	−5.42	−19.78	8.94	0.459

## Data Availability

Not applicable.

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
