# Peer review of "Self-Reported Medication Adherence Measured with Morisky Scales in Rare Disease Patients: A Systematic Review and Meta-Analysis"

_healthcare, 2023, doi:10.3390/healthcare11111609_

Round 1

Reviewer 1 Report

Thank you for the opportunity to review this article. The authors' goal is to provide a review regarding the level of adherence to medical therapy in patients with rare diseases. As well explained by the authors, this is a relevant topic that deserves to be treated by scientific research.

I congratulate the authors, I think some corrections are needed, in particular, inherent to English, but in my opinion, the article deserves to be published.

I have some comments:

-   In ABSTRACT, Background: I believe that the phrase "Presenting a serious development" should be linked to the previous period.

-   In ABSTRACT, Objectives: authors should unify "meta-analysis" and "meta-analysis," also in the text of the article there are both forms

-   In Figure.1: at the screening phase, the authors twice reported the elimination of some items as duplicates. I invite them to explain why the removal of duplicates was not done at the same time or to differentiate the two phases in the figure

-   In Figure.1: the authors did not include the number of studies included in the systematic review and meta-analysis. I invite them to correct the mistake

-   In Discussion: I agree with the authors' choice to describe the pathologies covered in the included articles. However, in some cases, the authors refer to other studies by citing the authors' names, while in others they do not. I encourage them to always name the author of the study because it would make the reading more fluent and comprehensible.

Author Response

----------------------------------------------------------------------------------------------------------

Reviewer#1

----------------------------------------------------------------------------------------------------------

Comment 1: I congratulate the authors, I think some corrections are needed, in particular, inherent to English, but in my opinion, the article deserves to be published.

Reply: English has been reviewed through the whole manuscript.

Comment 2:  In ABSTRACT, Background: I believe that the phrase "Presenting a serious development" should be linked to the previous period.

Reply: Both sentences have been linked and that has improved manuscript readability.

Comment 3: In ABSTRACT, Objectives: authors should unify "meta-analysis" and "meta-analysis," also in the text of the article there are both forms

Reply: The way of referring to meta-analysis is now unified in the manuscript.

Comment 4: In Figure.1: at the screening phase, the authors twice reported the elimination of some items as duplicates. I invite them to explain why the removal of duplicates was not done at the same time or to differentiate the two phases in the figure.

Reply: Thank you for your insightful comments regarding Figure 1. We agree with your suggestion and would like to clarify the differentiation of duplicate elimination in our manuscript.

During the first phase of duplicate removal, we eliminated exact duplicates - these are articles that were identified by more than one database during the literature search, which is a common occurrence when multiple databases are used.

On the other hand, during the "Reports assessed for eligibility" phase, we further removed duplicates in a more nuanced sense. Here, duplicates refer to multiple articles originating from the same research study. Often, one piece of research may yield multiple publications, for example, different aspects of the research might be published separately. However, they still pertain to the same study, so for the purpose of our analysis, they were considered duplicates.

To differentiate these in the flowchart, we suggest using the term "Research-based Duplicates" for the second phase. This terminology should clarify the distinct stages of duplicate removal in our screening process.

Comment 5:  In Figure.1: the authors did not include the number of studies included in the systematic review and meta-analysis. I invite them to correct the mistake

Reply: Thank you for pointing out the omission in Figure 1. It appears to have been a formatting oversight on our part, and we appreciate your attention to detail.

We have since revised the figure and included the number of studies included in the systematic review and meta-analysis. The revised figure now correctly represents the complete information about our research process.

Comment 6:  In Discussion: I agree with the authors' choice to describe the pathologies covered in the included articles. However, in some cases, the authors refer to other studies by citing the authors' names, while in others they do not. I encourage them to always name the author of the study because it would make the reading more fluent and comprehensible.

Reply: Thank you for this suggestion, we have followed your advice and authors’ names have been added through the whole discussion.

Reviewer 2 Report

o   In the abstract, the objective is not clear, it should be rewritten. In the abstract (line 12/13) ......... were identified through database searches and in other articles bibliography. What does it mean other articles bibliography? Do you want to say other sources such as google scholar? I do not understand it. Aged 0-83 years were included ......(Line 14) better to change it to age less than or equal to 83 years included.

o   The introduction is not clear and concise. It should be rewritten in clear, simple, and understandable language.

o   The Materials and Methods lack several points such as: 1) It was mentioned that one of the exclusion criteria was studies that were not a cross-sectional or longitudinal design. What about other kinds of studies? For example, retrospective studies, and randomized control trials? And others. You already mentioned that in the inclusion criteria as there is no restrictions apart from systematic reviews, and/or meta-analyses, qualitative and case studies. 2) in the subsection, Information sources and search strategy, you mentioned only PubMed, Scopus, Web of Science, and Cochrane Database of Systematic Reviews databases. What about others such as google scholar........ 3) Quality assessment was not carried out for the selected articles. For example, Newcastle-Ottawa Scales is used to assess the quality of selected articles. 4) The authors mentioned only prevalence/proportion bias. What about other sources of bias that may affect your study?

o   The results are clearly presented but some points should be addressed, for example, 1) In figure 1 (2 records) were excluded without mentioning the reason. Could you please mention the reason? 2) it would be nice if figure 1 be in one page. 3) in subsection 3.4.1, p<0.00 was indicated. I think it should be p<0.001. Please change if there are any.   

o   The discussion/conclusion is clear and concise. But a few points should be addressed, for example, you raised several points in the second paragraph of discussion without any evidence. Could you please mention some references? The last paragraph, which talks about the limitation of the study, should go up (above the conclusion paragraph).

Finally, I couldn’t find the supplementary materials. 

Author Response

Comment 1: In the abstract, the objective is not clear, it should be rewritten. In the abstract (line 12/13) ... were identified through database searches and in other articles bibliography. What does it mean other articles bibliography? Do you want to say other sources such as google scholar? I do not understand it. Aged 0-83 years were included ......(Line 14) better to change it to age less than or equal to 83 years included.

Reply: Thank you for all the wise suggestion regarding to the abstract content, all changes have been made accordingly.

Comment 2: The introduction is not clear and concise. It should be rewritten in clear, simple, and understandable language.

Reply: Thanks for the comment, substantial changes have been made in the introduction.

Comment 3: The Materials and Methods lack several points such as:

Comment 3. 1) It was mentioned that one of the exclusion criteria was studies that were not a cross-sectional or longitudinal design. What about other kinds of studies? For example, retrospective studies, and randomized control trials? And others. You already mentioned that in the inclusion criteria as there is no restrictions apart fromsystematic reviews, and/or meta-analyses, qualitative and case studies.

Reply: Thank you for your insightful comments regarding our inclusion and exclusion criteria. You've raised a valid point about the potential for confusion in our original text. Based on your comments, we have revised the relevant section of our manuscript for clarity.

Comment 3.2 in the subsection, Information sources and search strategy, you mentioned only PubMed, Scopus, Web of Science, and Cochrane Database of Systematic Reviews databases. What about others such as google scholar........

Reply: Thank you for your question about our choice of databases for the literature search. Our choice to utilize PubMed, Scopus, Web of Science, and the Cochrane Database of Systematic Reviews instead of Google Scholar was deliberate. We focused on these databases because they provide robust collections of peer-reviewed literature, ensuring study reliability and validity. Although Google Scholar includes a broad scope of literature, it also encompasses grey literature, which often lacks the rigorous peer-review process and standardized reporting that our review necessitated. Moreover, Google Scholar's search algorithms and inclusion criteria lack transparency, potentially compromising the replicability of our search strategy. Thus, to ensure the quality and reproducibility of our research, we chose to exclude it from our search.

Comment 3.3 Quality assessment was not carried out for the selected articles. For example, Newcastle-Ottawa Scales is used to assess the quality of selected articles. 

Reply: Thank you for your suggestion of using the Newcastle-Ottawa Scale for quality assessment of the selected articles in our study. We appreciate your thoughtful insights into our research methodology.

However, we would like to clarify that the tool we used for assessing the risk of bias, developed by Hoy et al. (2012), was specifically designed for prevalence studies, which aligns more closely with the type of studies included in our review. This comprehensive tool covers a broad range of potential biases, assessing 10 items related to both internal and external validity of the studies.

The Newcastle-Ottawa Scale, while valuable for assessing the quality of non-randomized studies such as cohort and case-control studies, may not be as well-suited for our research, which primarily involves prevalence studies. Thus, we believe that the use of Hoy et al.'s tool is more appropriate and sufficient for our study's objectives.

We appreciate your understanding and your keen attention to the methodological rigor of our research.

References:

- Hoy, D.; Brooks, P.; Woolf, A.; Blyth, F.; March, L.; Bain, C.; Baker, P.; Smith, E.; Buchbinder, R. Assessing Risk of Bias in Prevalence Studies: Modification of an Existing Tool and Evidence of Interrater Agreement. J. Clin. Epidemiol. 2012, 65, 934–939, doi:10.1016/j.jclinepi.2011.11.014.

Comment 3.4 The authors mentioned only prevalence/proportion bias. What about other sources of bias that may affect your study?

Reply: Thank you for your important query regarding the potential sources of bias in our study. We want to clarify that we have indeed taken into account a variety of potential biases through our use of the assessment tool by Hoy et al. (2012). This tool evaluates the risk of bias in prevalence studies across multiple dimensions, not just prevalence/proportion bias.

The tool encompasses a total of 10 items addressing both internal and external validity, including elements such as sample representativeness, sample size, non-respondents, data collection method, case definition, measurement tool validity and reliability, and appropriate statistical analysis. Each item was classified as "yes" (low risk) or "no" (high risk), with a corresponding score that helped categorize each study as "low risk of bias" (scores of 0-3), "moderate risk of bias" (scores of 4-6), or "high risk of bias" (scores of 7-9).

We apologize if our original wording in the manuscript may have led to some misunderstanding, suggesting we focused solely on prevalence/proportion bias. We appreciate you bringing this to our attention and allowing us to clarify this important aspect of our study design.

Comment 4: The results are clearly presented but some points should be addressed, for example, 1) Infigure 1 (2 records) were excluded without mentioning the reason. Could you please mention the reason?

Reply: Thank you for pointing out this oversight in our PRISMA flow diagram. We have now revised Figure 1 to include the reasons for the exclusion of the two records. These records were excluded after the title and abstract screening stage due to not meeting our inclusion criteria.

Comment 4.2: it would be nice if figure 1 be in one page.

Reply: Thank you for your suggestion regarding Figure 1. We have now revised Figure 1 accordingly to fit within a single page.

Comment 4.3: in subsection 3.4.1, p<0.00 was indicated. I think it should be p<0.001. Please change if there are any.   

Reply: Thank you for your attention to detail and pointing out the typo in subsection 3.4.1. We agree that p-values should be reported as p<0.001 instead of p<0.00. We have made the necessary corrections in the manuscript.

[PURA] Comment 5: The discussion/conclusion is clear and concise. But a few points should be addressed, for example, you raised several points in the second paragraph of discussion without any evidence. Could you please mention some references? The last paragraph, which talks about the limitation of the study, should go up (above the conclusion paragraph).

Reply: Thank you for this, some references have been added to the manuscript, and the paragraph related with limitations has changed its location.

Reviewer 3 Report

Title: Self-Reported Medication Adherence Measured with Morisky Scales in Rare Disease Patients: A Systematic Review and Meta-Analysis.

Reviewer Comments: 

In this systematic review authors tried to analyze the adherence to medication of the most common rare diseases. Adherence to treatment data was obtained using Morisky Medication Adherence Scale. Out of 54 studies only 18 studies were included in this systematic review and meta-analysis. Approximately 1600 participants, aged 0−83 years were included in this systematic review. The adherence to treatment data obtained from patients with rare diseases show drastic variability due to the various aspects involved in the greater or lesser applicability of the treatment.

1.    Figure 2 and 3 quality need to be improved.

2.    Change red color punctuation mark to black color at the end of 47 line.

3.    We understand that there is no gold standard method to assess medication adherence, however using different methods in combination might improve reliability.

4.    The authors should check the robustness of their prevalence estimates by conducting sensitive analyses using different transformation methods including arcsine transformation and logit transformation. Further, apart from pooling the results using the standard inverse variance method they should also use the Generalized Linear Mixed-Model (GLMM) method to pool their results. It would be interesting to see if the prevalence estimates differ significantly or not. Conducting these sensitivity analyses may also help reduce the increased heterogeneity observed in the pooled prevalence estimates.

5.    The PRISMA flow diagram (Figure. 1) is inaccurate. It does not show the total number of studies finally included in the systematic review.

6.    The authors observed significant heterogeneity in their pooled estimates. They should further explore the source of heterogeneity. Although the authors conducted a meta-regression analysis using mean age, but it is not sufficient. They should include more predictor variables in the meta-regression analysis such as publication year, study design, quality score etc. and any other relevant variable to further explore for the source of heterogeneity.

Author Response

Comment 1: Figure 2 and 3 quality need to be improved.

Reply: Thank you very much for your appreciation. The quality of figures 2 and 3 has been improved. The manuscript has been updated with the new images.

Comment 2: Change red color punctuation mark to black color at the end of 47 line.

Reply: Thank you for your attention to detail and for pointing out the typo. The manuscript has been updated.

Comment 3: We understand that there is no gold standard method to assess medication adherence, however using different methods in combination might improve reliability.

Reply: Thank you for this suggestion, we consider as well that, if available, different adherence measures could enrich manuscript content and results, however there are a few limitations that would make impossible to properly address the comment.

Rare diseases have a lack of medication resources available as it is stated in different resources to name a few: “Approximately 7,000 rare diseases have been identified. While each rare disease affects a small number of people, together rare diseases affect more than 25 million Americans. More than 90% of rare diseases have no FDA-approved treatment.” https://rarediseases.org/new-study-investigates-the-number-of-available-orphan-products-generics-and-biosimilars/

And even though there are several ways either objective (e.g., blood samples) or subjectives (e.g., questionnaires such as the medication adherence report scale).

Lam WY, Fresco P. Medication Adherence Measures: An Overview. Biomed Res Int. 2015;2015:217047. doi: 10.1155/2015/217047. Epub 2015 Oct 11. PMID: 26539470; PMCID: PMC4619779.

Morisky 4 and 8 is the more extended, since its short but with high values of sensitivity and specificity, making for the purpose of the present work the chosen option.

Comment 4: The authors should check the robustness of their prevalence estimates by conducting sensitive analyses using different transformation methods including arcsine transformation and logit transformation. Further, apart from pooling the results using the standard inverse variance method they should also use the Generalized Linear Mixed-Model (GLMM) method to pool their results. It would be interesting to see if the prevalence estimates differ significantly or not. Conducting these sensitivity analyses may also help reduce the increased heterogeneity observed in the pooled prevalence estimates.

Reply: Thank you very much for your comments. Other transformations have been performed to further enrich the manuscript, as you suggested. The results of the new transformations can be found on lines 218-221.

Comment 5: The PRISMA flow diagram (Figure. 1) is inaccurate. It does not show the total number of studies finally included in the systematic review.

Reply: Thank you for pointing out the omission in our PRISMA flow diagram. We appreciate your careful review. We have now updated Figure 1 to accurately reflect the total number of studies included in the final systematic review.

Comment 6: The authors observed significant heterogeneity in their pooled estimates. They should further explore the source of heterogeneity. Although the authors conducted a meta-regression analysis using mean age, but it is not sufficient. They should include more predictor variables in the meta-regression analysis such as publication year, study design, quality score etc. and any other relevant variable to further explore for the source of heterogeneity.

Reply: We appreciate your suggestions regarding the exploration of heterogeneity in our meta-regression analysis. We have carefully considered your recommendations and have made significant updates to our model by including additional predictor variables. These additions can be found on lines 249-251 of the revised manuscript.

In addition, we have added a new table (Table 2) to the manuscript, designed specifically to present the results of the extended regression meta-analysis. This table provides an overview of the findings and allows a deeper understanding of the relationships between predictor variables and sources of heterogeneity.

Round 2

Reviewer 2 Report

All the comments raised are addressed well